

# An open platform for Aerosol InfraRed Spectroscopy analysis – AIRSpec

Matteo Reggente, Rudolf Höhn, and Satoshi Takahama

ENAC/IIE Swiss Federal Institute of Technology Lausanne (EPFL), Lausanne, Switzerland

*Correspondence to:* Satoshi Takahama (satoshi.takahama@epfl.ch)

**Abstract.**

AIRSpec is a platform consisting of several chemometric packages developed for analysis of Fourier Transform Infrared (FTIR) spectra of atmospheric aerosols. The packages are accessible through a browser-based interface, which also generates the necessary input files based on user interactions for provenance management and subsequent use with a command-line interface. The current implementation includes the task of baseline correction, organic functional group (FG) analysis, and multivariate calibration for any analyte with absorption in the mid-infrared. The baseline correction module uses smoothing splines to correct the drift of the baseline of ambient aerosol spectra given the variability in both environmental mixture composition and substrates. The FG analysis is performed by fitting individual Gaussian line shapes for alcohol COH, carboxylic COOH, alkane CH, carbonyl CO and amine NH for each spectrum. The multivariate calibration model uses the spectra to estimate the concentration of relevant target variables (e.g., organic or elemental carbon) measured with different reference instruments. In each of these analyses, AIRSpec receives spectra and user choices on parameters for model computation; input files with parameters that can later be used with a command-line interface for batch computation are returned together with diagnostic figures and tables in text format. AIRSpec is built using the open source software consisting of R and Shiny and is released under the GNU Public License v3. Users can download, modify, and extend the package, or access its functionality through the web application (http://airspec.epfl.ch) hosted at the École Polytechnique Fédérale de Lausanne (EPFL). AIRSpec provides a unified framework by which different chemometric techniques can be shared and accessed, and its underlying suite of packages provides the basic functionality for extending the platform with new types of analyses. For example, basic functionality includes operations for populating and accessing spectra residing in in-memory arrays or relational databases, input and output of spectra and results of computation, and user interface development. This paper describes the modular architecture and provides examples of the implemented packages using the spectra of aerosol samples collected on PM$_{2.5}$ Polytetrafluoroethylene (Teflon) filters.

## 1 Introduction

Atmospheric particulate matter (PM) has been associated with increased morbidity and mortality (Janssen et al., 2011; Anderson et al., 2012), reduced visibility (Watson, 2002; Hand et al., 2012), and is one of the least understood components of the climate system (Yu et al., 2006; Bond et al., 2013). Chemical characterization of PM is paramount for understanding its source



origins and properties such as the extent of oxidation and hygroscopicity, which determine its eventual fate. Fourier transform infrared spectroscopy (FTIR) is a technique that measures the absorption spectrum of molecules that can be related to its underlying structure. A particular challenge of atmospheric PM is that it is composed of a complex mixture of thousands of different molecules that vary in structure and physicochemical properties (Seinfeld and Pandis, 2016), which poses significant chal-
lenges for chemical characterization by any method or suite of methods. For FTIR, this complexity can lead to broadened and overlapping absorption bands, with significant scattering or absorption contributions from the substrate additionally impeding consistent interpretation across users.

Nonetheless, FTIR has provided chemically-informative and cost-effective means for PM characterization in intensive measurements campaigns (e.g., Maria et al., 2002, 2003b; Russell, 2003) and monitoring network samples [e.g., Interagency Mon-
itoring of PROtected Visual Environments (IMPROVE) network and the Chemical Speciation Network/Speciation Trends Network (CSN/STN) in the USA]. For example, inorganic salts (Cunningham et al., 1974; Allen et al., 1994), dust (Foster and Walker, 1984), organic functional groups (Allen et al., 1994; Maria et al., 2002, 2003b; Chen et al., 2016; Coury and Dillner, 2008; Takahama et al., 2013, 2016; Faber et al., 2017), and carbonaceous content (Dillner and Takahama, 2015a, b; Reggente et al., 2016) have been estimated by calibration models developed for FTIR spectra. Spectra clustering and factor analysis have
been used to estimate source contributions from fossil fuels, vegetation, marine environements, and biomass burning (Russell et al., 2009a; Liu et al., 2009; Russell et al., 2011; Takahama et al., 2011; Frossard et al., 2014).

In this paper, we present a framework, AIRSpec (Aerosol InfraRed Spectroscopy), that simplifies the writing and deployment of chemometric software packages for FTIR spectra processing and analysis for atmospheric aerosols and harmonizes results across users. The objective of this program is not to re-implement general purpose spectroscopic tools (e.g., Bruker OPUS) or
chemometrics software (e.g., CAMO Unscrambler), but to provide a platform for sharing code specifically developed and used for the analysis of FTIR spectra of atmospheric aerosol samples.

AIRSpec is built using the open source R statistical environment (R Core Team, 2016) and the Shiny web application framework (Chang et al., 2016), which permits the user to install the software locally or access its functionality through the web at *http://airspec.epfl.ch* (hosted at the École Polytechnique Fédérale de Lausanne, EPFL). AIRSpec provides a common
data object that facilitates storage of and operations on spectra using in-memory arrays or relational databases, upon which chemometric packages that encode common decisions made for processing of spectra are built. A user interface to facilitate exploratory work consolidates the various packages and allows information to be passed among them; while power users can extend functionality or carry out batch analyses using scripts. Extensive documentation and template files provided in the demo folder of the package are provided.
In the current version, AIRSpec implements only chemometric packages of algorithm and methods already published. The available chemometric package are: (i) the spectra baseline correction algorithm proposed by Kuzmiakova et al. (2016) to counteract the drift of the baseline of ambient aerosol spectra given the variability in both environmental mixture composition and substrates; (ii) the multiple peak-fitting algorithm proposed by Takahama et al. (2013) to perform FG analysis for alcohol COH, carboxylic COOH, aliphatic CH, carbonyl CO and amine NH; and (iii) multivariate regression and calibration approach
described by Dillner and Takahama (2015a, b); Reggente et al. (2016). Moreover, AIRSpec uses interactive plots to facilitate the



exploratory work as described in Sections 3.1.2 and 3.3.2, and, because of its modular structure, new chemometric packages can be easily added. Therefore, the objective of AIRSpec is also to provide a platform to facilitate the utilization of chemometric packages – that can be integrated with new ones – for the analysis of FTIR spectra of atmospheric aerosol samples. The AIRSpec package also includes a demo folder where the interested user can find instructions on how to add a new chemometric

package or script in a few steps.

We explain the modular architecture (Section 2) and provide further details on the implementation of the chemometric packages (Section 3). We conclude with a summary and outlook (Section 4).

## 2    Architecture and workflow

A diagram of the AIRSpec framework and associated chemometric packages is shown in Fig. 1. Each package is a collection

of functions and scripts for accomplishing specific computations or visualizations (e.g., baseline correction); users can access available functionality through R scripts run interactively or directly through the command-line interface (CLI). Alternatively, the AIRSpec framework provides a server layer to connect users with existing scripts through Shiny Modules (henceforth referred to with capital *M*) that execute common tasks via CLI, and permit each Module to use outputs of other Modules. (Formally, Shiny Modules are self-contained pairs of interface and server definitions that comprise an app that can be embedded

within larger apps.) New packages (denoted with dotted boxes and arrows in Fig. 1) can furthermore be incorporated into this modular framework.

### 2.1    Server

The server layer more generally consists of actions to be performed in response to user interaction with an interface. In the AIRSpec framework, the server typically invokes pre-defined system calls to the operating system to run package scripts

through the CLI; accepting the name of a single parameter file as input. The directory of the input file also becomes the destination for all output files, so that the results can be stored together with their input parameters. The input file defines all necessary information, including location and name of spectra files, and parameters of functions used by the script. The input parameter file is specified in JSON (Javascript Object Notation) format, which is a standard hierarchical data exchange format that is human-readable and editable. The scripts and interface-generated input files are stored on the server side and, on account

of security, the user cannot edit them directly through the web interface. Details on implementation as they pertain specifically to the R language are provided in the package help files, but a basic overview is presented here.

### 2.2    User interface (UI)

The AIRSpec user interface (UI) module provides a mechanism for generation of input parameter files and execution of predefined scripts without the requirement for using the CLI directly. The landing page (homepage) provides introductory

information regarding the tool and possibility to download template files on which users can base their input files for the chemometric packages.



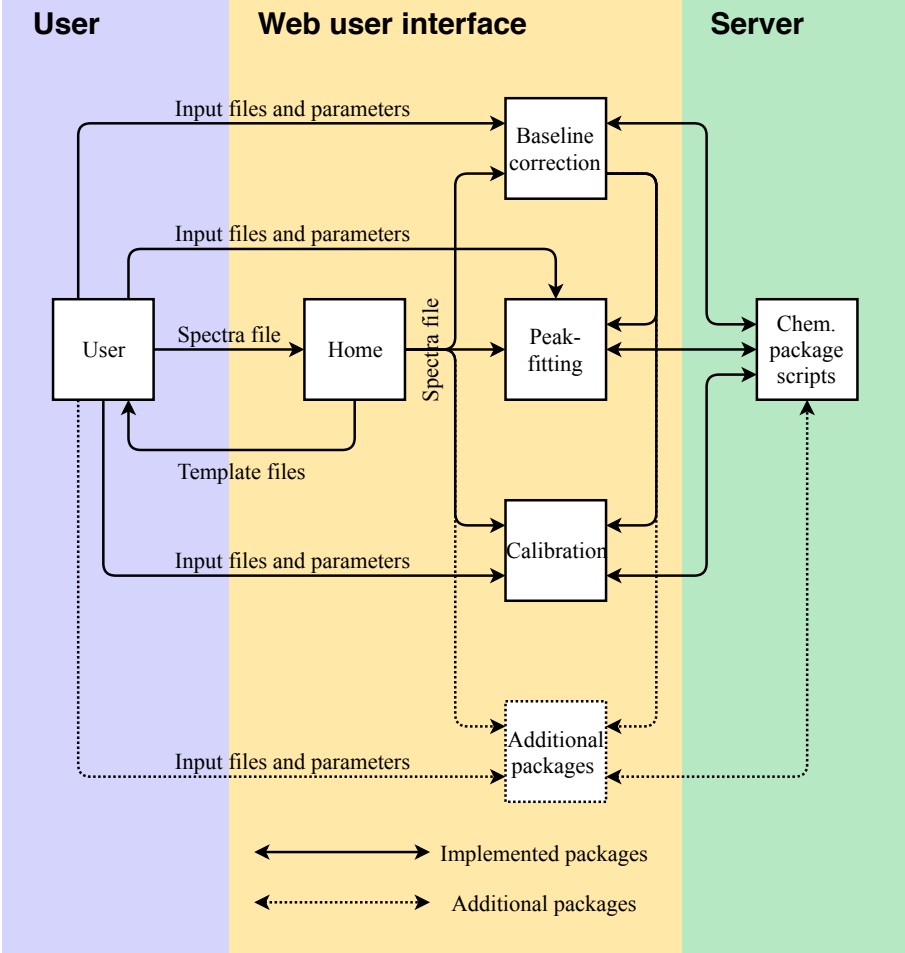

**Figure 1.** AIRSpec architecture diagram and workflow.

**Homepage.** On this homepage, the user typically uploads the spectra file, after which 100 random spectra are plotted for confirmation (Fig. 2). The user can then access the available chemometric packages through the additional navigation tabs. Each tab corresponds to a chemometric package, and the user can upload required, and optional files or change default inputs (Table 1 summarizes the list of inputs for each tab). In each package tab, after the computation has concluded, the download

5    button appears, and the user can retrieve an archive (zip) file containing results and input parameter files. This folder contains all the information about the analysis.

**Tabs.** In its most straightforward configuration, a chemometric package deployed on AIRSpec (solid boxes and arrows in Fig. 3) requires a script which accepts an input parameter file (JSON format) and four nested Shiny Modules, which are described in turn. The user input handler module (solid red lines) writes the user inputs to the input parameter file stored in the

10    server. The program wrapper (solid blue lines) runs the scripts of the chemometric package, passing the location of the input



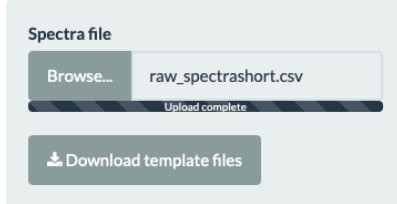

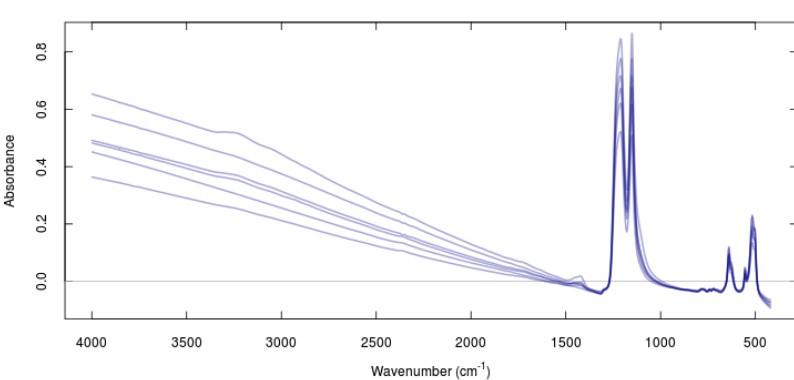

**Figure 2.** AIRSpec homepage.



**Table 1.** List of inputs for each chemometric package.

| Tab | Mandatory | With default value | Optional |
|---|---|---|---|
| Home | Spectra file | | |
| Baseline correction | | Sequence of EDF values | Sample list file |
| | | Checkbox for non-negativity constraint | |
| Peak fitting | | Spectra type | Sample list file |
| | | Bond sequence | |
| Calibration | Response file | Spectra type | Wavenumbers |
| | Case file | Variables | Minimum detection limit |
| | | Max number of PLS components | |
| | | Number of CV segments for CV | |
| | | PLS method | |
| | | PLS type | |
| | | Parameters optimization | |
| | | Segment type | |
| | | Order response variable | |

parameter file and saving the output files in the same folder of the input parameter file. The results and diagnostic renderer (solid orange lines) plot the output. Finally, the download handler (solid green lines) prepares and archives the input parameter file and source code output in a downloadable file.

Except for the specifications of the user input handler (e.g., input files, parameters), all the modules are general and used

5 by each chemometric package. Therefore, to incorporate functionalities of a new package, the user only needs to prepare a the package scripts and the Module containing specifications of the permitted user inputs. AIRSpec additionally uses nested modules to handle ad-hoc features such as dynamic input handling or ad-hoc reactive plots. Again, because of the modular structure, the addition of ad-hoc features does not require the changing of the necessary modules.

## 2.3 Relational database integration

10 AIRSpec uses the APRLspec package to handle its primary data object (spectra matrix) and essential operations (e.g., selecting and merging of spectra), and I/O (input/output) functions. The primary data object defines a spectrum as a row matrix with wavenumber attributes, which is a standard data structure for multivariate analysis. A new class for this type of data is defined so that operations can manipulate spectra columns and wavenumber attributes simultaneously to reduce errors of mismatched dimensions.





**Figure 3.** Package diagram and workflow.





With an increasing number of spectra – for example in Reggente et al. (2016) the authors used thousand of spectra for their analysis – efficiency may be gained by building a spectra archive in a central, relational database. In this way, all spectra did not need to be loaded into virtual memory at once (which can be a limiting factor in analyses) but retrieved selectively from the database as necessary. The APRLspec package provides functions for accessing and storing spectra in a SQL (Structured

Query Language) relational database using the similar syntax used for spectra matrices residing in virtual memory. The current implementation has been tested and used in the baseline correction package with SQLite, an embedded relational database.

## 3    Chemometric packages

In this Section, we briefly summarize the chemometric packages developed for the analysis of atmospheric samples and currently integrated into AIRSpec: baseline correction (APRLssb R package), peak-fitting (APRLmpf R package), and multivari-

ate calibration (APRLmvr R package). For each package, we first introduce the methods and then we illustrate the analysis that the user can obtain using AIRSpec. The results are based on aerosols samples collected on PM$_{2.5}$ Teflon filters. The samples were collected at IMPROVE sites on every third day in 2011. The sample collection methods and results have been already discussed in previous works (Kuzmiakova et al., 2016; Takahama et al., 2016; Takahama and Ruggeri, 2017; Dillner and Takahama, 2015a, b; Reggente et al., 2016, 2018), and here we present some of the already published results, to give an example of

what an AIRSpec user can obtain from it.

In the baseline package (Section 3.1.2), we first baseline correct the spectra matrix, discussing the EDF parameters selection. In Section 3.2.2, we use the baseline corrected spectra of two sample (one collected in the urban site Phoenix, Arizona, and one collected in the rural location St. Marks (Florida)) to quantify the FGs, the OM, and OM/OC. In Section 3.3.2, we use the baseline corrected spectra and collocated measurements of OC and EC (measured with thermal-optical methods), to calibrate

the spectra to estimate unseen OC and EC concentrations.

### 3.1    Smoothing spline baseline correction

#### 3.1.1    Background

Scattering by particles and the substrate used for their collection creates an interfering signal that hinders quantitative analysis of absorption spectra using the Beer-Lambert law. The subtraction of the spectrum filter before sample exposure (blank) from

the aerosol spectrum provides an insufficient remedy. A smoothing spline (Reinsch, 1967) is used to model the baseline by regressing onto background regions (where no analyte absorption is expected) and interpolating through the analyte region (Kuzmiakova et al., 2016). The spectrum is divided into multiple regions so that optimal baseline correction parameters can be obtained for each segment. For each spectrum and each segment, the baseline is obtained for a given penalty smoothness penalty $\lambda$ by minimizing the objective function

$$\min_{\hat{f}} \sum_{j=1}^{N} w_j [y_j - \hat{f}(\tilde{\nu}_j)]^2 + \lambda \int [\hat{f}''(t)]^2 dt \,, \tag{1}$$





$y_j$ and $\hat{f}(\tilde{\nu}_j)$ are observed and fitted absorbances at wavenumber $j$, respectively, and $\lambda$ is a smoothing penalty. The first term represents the fidelity of fitted values to the background absorbance, and the regularization term imposes rigidity on $\hat{f}(\tilde{\nu}_j)$. The baseline is approximated by a natural cubic spline basis $\hat{f}(\tilde{\nu}) = \sum_{i=1}^{N} B_i(\tilde{\nu})\alpha_i$ with coefficient $\alpha$ for each basis function. $w$ is the weight at wavenumber $j$, which we define as:

$$w_j = \begin{cases} 0 & \text{if } \tilde{\nu}_j \in \mathcal{W}_A \text{ (analyte region)} \\ 1 & \text{if } \tilde{\nu}_j \in \mathcal{W}_B \text{ (background region)}. \end{cases} \tag{2}$$

$\lambda$ has a monotonic relationship with the effective degrees of freedom (EDF) of the smoothing spline fit (Hastie and Tibshirani, 1990; Green and Silverman, 1993), for which candidate values can be more easily understood through analogy to that of projection matrices commonly used in regression more generally. Letting $\boldsymbol{f} = [f(\tilde{\nu}_j)]$, $\mathbf{W} = \text{diag}(w_j)$, $\boldsymbol{\Omega}_N = [\int B_i''(t)B_j''(t)dt]$, and $\mathbf{K} = (\mathbf{B}^T)^{-1} \boldsymbol{\Omega}_N \mathbf{B}^{-1}$ with $\mathbf{B} = [B_i(\tilde{\nu}_j)]$, eq. 1 can be written as:

$$\min_{\hat{\boldsymbol{f}}} (\boldsymbol{y} - \boldsymbol{f})^T \mathbf{W} (\boldsymbol{y} - \boldsymbol{f}) + \lambda \boldsymbol{f}^T \mathbf{K} \boldsymbol{f},$$

for which the solution is given as a transformed version of the original spectrum:

$$\hat{\boldsymbol{f}} = \mathbf{B}\hat{\boldsymbol{\alpha}} = \mathbf{B} \left( \mathbf{B}^T \mathbf{W} \mathbf{B} + \lambda \boldsymbol{\Omega}_N \right)^{-1} \mathbf{B}^T \mathbf{W} \boldsymbol{y} = (\mathbf{W} + \lambda \mathbf{K})^{-1} \mathbf{W} \boldsymbol{y} = \mathbf{S}_\lambda \boldsymbol{y}. \tag{3}$$

For a given value of $\lambda$, the corresponding $\text{EDF}_\lambda = \text{Tr}(\mathbf{S}_\lambda)$ is defined by the trace of the smoother matrix $\mathbf{S}_\lambda$, which in turn depends on the penalty matrix $\mathbf{K}$ and the regression weights $\mathbf{W}$ (eq. 3). EDF replaces $\lambda$ and its value effectively determines the rigidness of the resulting baseline (EDF = 2 is a straight line) and must be selected by the user. For a candidate set of EDF values, the negative absorbance fraction (NAF) is used to evaluate whether a physically realistic spectrum is obtained. NAF represents the contribution of negative analyte absorbance, $\|\boldsymbol{a}_{A-}\|_1$, to the total analyte absorbance, $\|\boldsymbol{a}_A\|_1$:

$$\text{NAF} = \frac{\|\boldsymbol{a}_{A-}\|_1}{\|\boldsymbol{a}_A\|_1} \times 100\% \tag{4}$$

where $\|\cdot\|_1$ denotes the 1-norm magnitude of a vector (summation of all absolute values of vector elements). NAF is calculated across the entire wavenumber range in the analyte region of in a given segment.

In our current implementation, we provide $\mathcal{W}_B$ for two segments (4000–1820 $\text{cm}^{-1}$ and 2000–1500 $\text{cm}^{-1}$) in the high-frequency region ($>1500\,\text{cm}^{-1}$) of the spectrum where stretching and bending modes of bonds in carbon-hydrogen, carboxylic acid, carbonyl, hydroxyl, and amine groups are present (Maria et al., 2003a). This region has been used extensively for functional group analysis (Russell et al., 2011; Takahama et al., 2013) and estimation of carbon content (Dillner and Takahama, 2015a, b). The values of $\mathcal{W}_B$ provided undergo iterative adjustment by the algorithm to further avoid negative regions, and by this approach have generated suitable baselines for ambient PM samples (Kuzmiakova et al., 2016). However, for use with laboratory samples with specific absorption patterns and low concentrations may require additional adjustment.

### 3.1.2 Implementation

**Browser Interface**. In the baseline correction tab (Fig. 4), the user can upload an optional file containing the list of samples to process; AIRSpec will otherwise baseline correct all the samples present in the uploaded spectra file. A sequence of EDF





values for which different baselines are computed and compared are provided by default but can be edited by the user. When interpolating over a wide region with substantial curvature, negative regions may arise for any EDF if the degree of curvature is not well captured by the pre-specified background regions. AIRSpec provides an option to impose a non-negativity constraint, in which interior points with substantially negative values are iteratively added to $\mathcal{W}_B$ during the fitting process. This heuristic

approach provides a lower-bound envelope of the actual absorbance that is discernable from the spectrum. By clicking the `Compute` button, AIRSpec will execute prepared scripts, passing as input a file with the desired configuration generated from user choices. At the end of the computation, AIRSpec plots the spectra of the baseline-corrected samples using the EDF parameter which minimize the median of the NAF (Kuzmiakova et al., 2016). The EDF value can be changed by the user a posteriori, after which the plot will automatically update. A list of the output files can be found in Table 2 and in the dedicated

wiki tab in the web interface.

   **Example output**. Panel (a) of Fig. 5 shows the unprocessed and the baseline corrected spectra of ambient samples collected on PTFE filters.  The scatter from PTFE fibers results in a sloping baseline that masks the absorbance bands and therefore makes challenging to extract the analyte contributions (unprocessed spectra). Moreover, the baseline is not the same for each spectrum on account of fiber orientation and stretching. These variances make standardized baseline preprocessing methods

(e.g., blanks spectra or pre-scan subtraction) insufficient. A detailed description of these ambient samples can be found in the companion article (Reggente et al., 2018)).

   In the baseline corrected spectra (Fig. 5a) the features due to the analyte are more evident than the previous case. For example, the amine NH and carbonyl CO peaks (around 1600 and 1700 $\mathrm{cm}^{-1}$ respectively) are easily distinguishable, and it is possible to notice that there are filters with different amounts of these two FGs. Similarly, the sharp peaks around 2800 and

2900 $\mathrm{cm}^{-1}$ are due to the presence of alkane CH, and the two broad peaks around 3100 and 3300 $\mathrm{cm}^{-1}$ are due to the presence of ammonium NH, alcohol COH, carboxylic COOH and to lower extent to aromatic CH and alkene CH.

   The baseline corrected spectra have been computed using the EDF values that minimize the median of the NAF metric. We computed the baseline for different EDF values (2, 4, 6, 8, 10, 12 – the user can change these values), and the EDF values selected are 2 and 8 for segment 1 and 2 respectively (red dots in  Fig. 5b). The AIRSpec user can check these diagnostic

plots in the parameter selection tab, and change the proposed EDF values using the input field provided in the web interface. Moreover, the user, using the implemented interactive plot, can immediately see the changes in the spectrum shapes when changing the edf parameters, facilitating thus the exploratory work and parameter selection task.

## 3.2   Peak fitting

### 3.2.1   Background

The peak fitting package implements the physically-based approach proposed by  Alsberg et al. (1997), and it aims at quantifying the FG abundance of aerosol samples collected on PTFE filters. This method decomposes each spectrum in Gaussian peaks or lineshapes which represent the absorption profile of individual bonds. The FG abundances are estimated by relating absorption peaks or lineshapes with their molar absorption coefficients as described by Takahama et al. (2013).



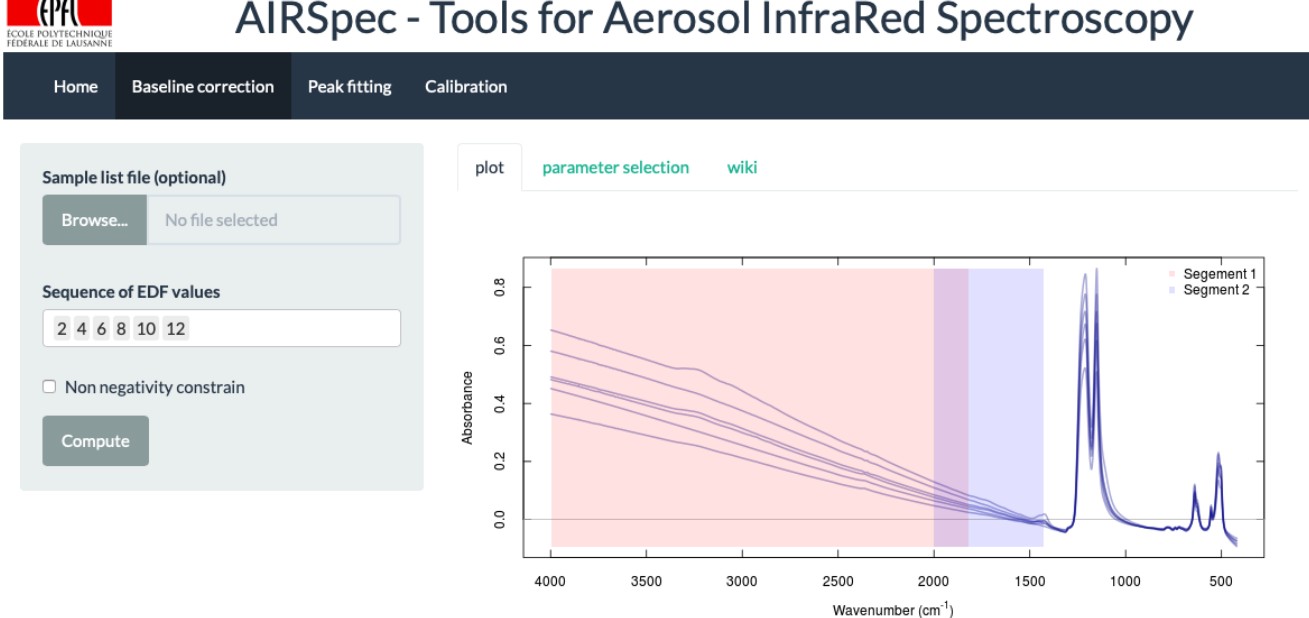

**Figure 4.** AIRSpec baseline correction package tab.

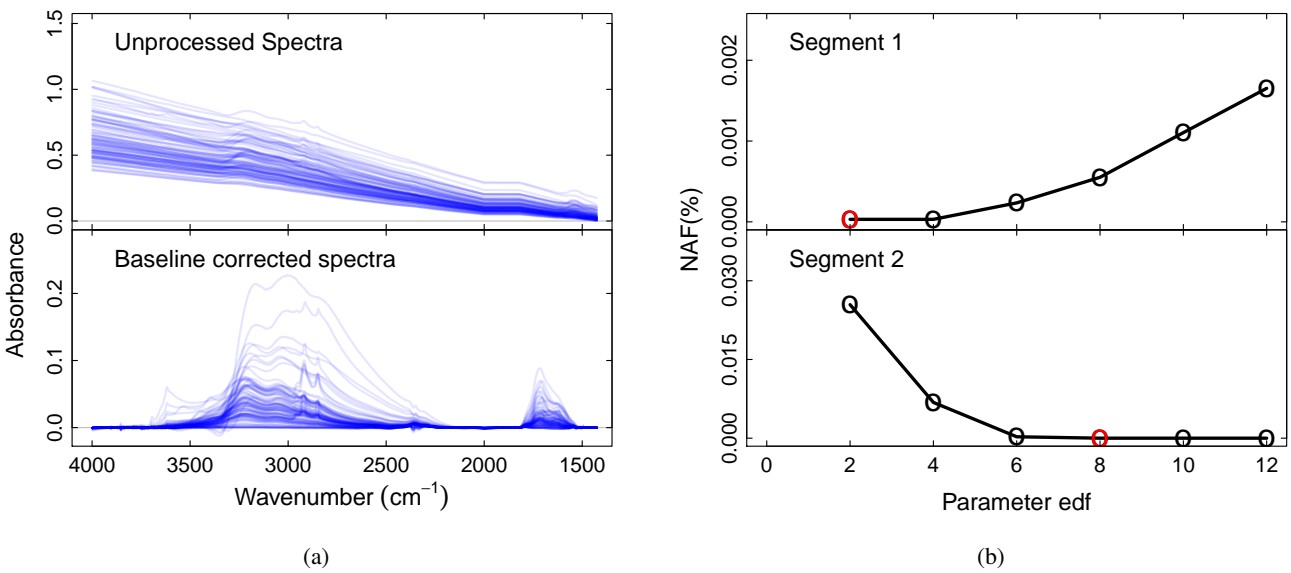

**Figure 5.** Baseline correction package outputs. (a) From top to down, the panels show the unprocessed and the baseline corrected spectra of ambient samples collected on Teflon filters. A detailed description of these ambient samples can be found in the companion article (Reggente et al., 2018)). (b) The EDF selection criteria for segment one and two. The red circles represent the EDF values that minimize the median of the selection criteria NAF.





**Table 2.** List of outputs of the Baseline correction chemometric package. {segment} denotes the segment label.

| Category | File | Notes |
|---|---|---|
| | | **Baseline correction package outputs** |
| | | |
| R package input files | `input_ssb_user.json` | Input parameters used in the computation of the baseline correction. |
| | `ssb_selected.json` | Selected EDF parameters and stitching method. |
| Package outputs | `{segment}_baseline.rds` | Baseline matrix of each segment. Each row represents a single baseline, and the sample names are provided in the rownames attribute; wavenumbers are saved in the "wavenum" attribute. |
| | `{segment}_baselinedb.sqlite` | SQL databases that contain baselines generated for each spectrum and parameter, and additional parameters if they are estimated internally. |
| | `{segment}_spec.rds` | Baseline-corrected absorbance spectra matrix of each segment. Each row represents a single spectrum, and the sample names are provided in the rownames attribute; wavenumbers are saved in the "wavenum" attribute. |
| | `spectra_baselined.csv` `spectra_baselined.rds` | Baseline-corrected absorbance spectra matrix of the stitched segments. In the .csv file, the first column contains the wavenumber values and the spectra in each of the subsequent columns (with sample names as column labels). In the .rds file, each row represents a single spectrum, and the sample names are provided in the rownames attribute; wavenumbers are saved in the "wavenum" attribute. Wavenumbers are saved in the "wavenum" attribute. |
| Summary files | `{segment}_agg_criteria_table.csv` | Summary statistics of the aggregated NAF and aggregated total normalized absolute blank absorbance (if blank samples are provided). |
| | `{segment}_baseline_param.csv` | Table with wavenumber bounds of the computed smoothing spline and the effective EDF of each baseline-corrected spectra. |
| | `{segment}_blankabs.csv` | Table with the aggregated total normalized absolute blank absorbance for each sample and EDF (only if blanks are provided). |
| | `{segment}_naf.csv` | Table with the NAF values for each sample and EDF. |
| Figures | `spectra_plots.pdf` | Spectra plots for each sample. Different color lines refer to spectra computed with different EDF. |
| | `{segment}_agg_criteria.pdf` | Figures of the summary statistics of the aggregated NAF and aggregated total normalized absolute blank absorbance (if blank samples are provided). |





The analysis consists of three parts: i) estimation of molar abundances of bonds from peak areas, ii) reapportionment of bonds to functional groups, iii) estimation of atomic abundances from functional group abundances (from which OM, OM/OC, O/C are derived). In the following equations, $s$ denotes a lineshape function defined over wavenumbers $\tilde{\nu}_j$s, and a set of peak parameters $\theta_{ik}$ for sample $i$ for bond $k$. The integrated absorbance (peak area) together with a molar absorption coefficient and for the bond is used to estimate the number of moles of bond per unit area of sample (denoted by $n$):

$$n_{ik} = \bar{\varepsilon}_k \Delta\tilde{\nu} \sum_j q_j s(\tilde{\nu}_j, \theta_{ik}) \tag{5}$$

$$x_{ij} = \sum_k s(\tilde{\nu}_j, \theta_{ik}) + e_{ij} \, . \tag{6}$$

$q_j$s are quadrature coefficients for numerical integration. For a Gaussian lineshape, $\theta_{ik}$ may correspond to any number of relevant amplitude, location, and width parameters for each bond, and an analytical solution exists for its integral. For fixed lineshapes (e.g., cCOH), the peak parameter corresponds to a scaling coefficient. Current default values for molar absorption coefficients $\varepsilon_k$ are those reported by Russell et al. (2009a) and Takahama et al. (2013).

The total carbonyl (tCO) quantified by peak fitting can include contributions from carboxyl, ketone, ester, and aldehyde carbonyl because of their proximity in absorption bands that are difficult to resolve in environmental samples (Takahama et al., 2013; Reggente et al., 2018). Therefore, we apportion tCO to acid (COOH) and non-acid CO (naCO):

$$(n_{\mathrm{COOH}}, n_{\mathrm{naCO}}) = \begin{cases} (n_{\mathrm{COOH}}, n_{\mathrm{tCO}} - n_{\mathrm{COOH}}), & \text{if } n_{\mathrm{tCO}} > n_{\mathrm{COOH}} \\ (n_{\mathrm{COOH}}, 0), & \text{otherwise.} \end{cases} \tag{7}$$

Using this convention, we estimate apportionment on an individual sample basis rather than in aggregate as described by Takahama et al. (2013). We compute the OC mass, OM/OC and O/C ratios by the constituent atomic molar abundance $n_a$, which is calculated from moles $n_k$ for FG $k$ through a coefficient $\lambda_{ak}$ such that $n_a = \lambda_{ak} n_k$. The value of $\lambda_{ak}$ are set according to the bonding configuration proposed by Takahama and Ruggeri (2017).

$$\begin{pmatrix} n_{\mathrm{C}} \\ n_{\mathrm{O}} \\ n_{\mathrm{H}} \end{pmatrix} = \begin{pmatrix} 0.5 & 1 & 0.5 & 1 & 0.25 \\ 1 & 2 & 0 & 1 & 0 \\ 1 & 1 & 1 & 0 & 1 \end{pmatrix} \begin{pmatrix} n_{\mathrm{aCOH}} \\ n_{\mathrm{COOH}} \\ n_{\mathrm{aCH}} \\ n_{\mathrm{naCO}} \\ n_{\mathrm{NH}} \end{pmatrix} \tag{8}$$

In contrast to previous studies (Russell, 2003; Ruthenburg et al., 2014), Takahama and Ruggeri (2017) proposed the value of 0.5 for $\lambda_{\mathrm{C \cdot aCOH}}$, which is consistent with past work by Russell et al. (2009b). A value of 0.5 corresponds to the assumption that the carbon shares an aCOH bond with a single aCH bond, whereas a value of 0 corresponds to the assumption of a terminal saturated carbon in which it is accounted for by two aCH bonds.




### 3.2.2  Implementation

**Browser Interface**. In the peak-fitting tab (Fig. 6), the user can choose to use the uploaded spectra matrix file (uploaded in home tab) or, if already computed, the baseline corrected spectra matrix (output of the baseline correction tab). Moreover, the user can choose the sequence of FGs (default provided) to be fitted, and the samples to process (as default AIRSpec will peak-fit all the samples present in the spectra matrix). By clicking the `Compute` button, AIRSpec will execute prepared scripts, passing as input a file with the desired configuration. At the end of the computation, AIRSpec plots each spectrum with the fitted peaks . Moreover, the user can download a table containing the FG distribution in organic mass (OM) and the OM/OC ratio (for each sample) A list of the available files can be found in Table 3 and in the dedicated wiki tab in the web interface.

**Example output**. We fit individual Gaussian line shapes to obtain the abundance (in moles) of alcohol COH, carboxylic COOH, alkane CH, total carbonyl CO and amine NH from the baseline corrected spectra.  Figure 7a shows the fitted peaks and the spectra of the sample collected in one rural and one urban site. According to the Beer-Lambert law (variations of FGs abundance are linearly dependent to the absorbance), from a visual comparison of the two samples, we can note that the urban sample is characterized by greater abundance of amine NH (orange peaks around 1600 $cm^{-1}$) and total carbonyl CO (dark green peaks around 1700 $cm^{-1}$) FGs. Moreover, the urban sample shows greater abundances of carboxylic COOH (light green bimodal line) and alkane CH (sharps blue peaks around 2800 and 2900 $cm^{-1}$). The rural site, instead, is characterized by a greater abundance of ammonium NH (dark orange bimodal line).

We used the method described in Takahama and Ruggeri (2017) to quantify the FGs, OM, and OM/OC. If the AIRSpec user provides the collecting area of the filter used to trap the aerosol (3.53 $cm^2$ in the example of Fig. 7a) and the nominal flow rate (22.8 L $min^{-1}$ in the example of Fig. 7), AIRSpec uses this information to convert the area of the fitted peaks to the FG concentrations (in $\mu g/m^3$).  AIRSpec returns the OM computed in $\mu moles/cm^2$. In Fig. 7b , the first bar plot compares the FG distribution in the OM for the two samples. First, we note that OM is higher in the urban site, and the alkane CH accounts the 38% and 57% of the total OM in the rural and urban site respectively. These differences are mainly due to the higher impact of anthropogenic sources in the urban site. Moreover the significant contributions of alcohol (21% and 14% in the rural and urban samples respectively) and carboxylic acid (34% and 23% respectively) exemplify the influence of processed aerosol from surrounding regions affecting the $PM_{2.5}$. Moreover, the OM/OC mass and O/C ratios (second and third bar plots in Fig. 7b  respectively) are higher in the rural site than the urban (2.08 and 1.72 respectively). This result is in agreement with measurements by GC-MS and AMS (Turpin and Lim, 2001; Aiken et al., 2008) and can be explained by the condensation of functionalized molecules (Ziemann, 2005; Kroll and Seinfeld, 2008) and heterogeneous reactions (Smith et al., 2009; Lim et al., 2010) which lead to chemical aging.



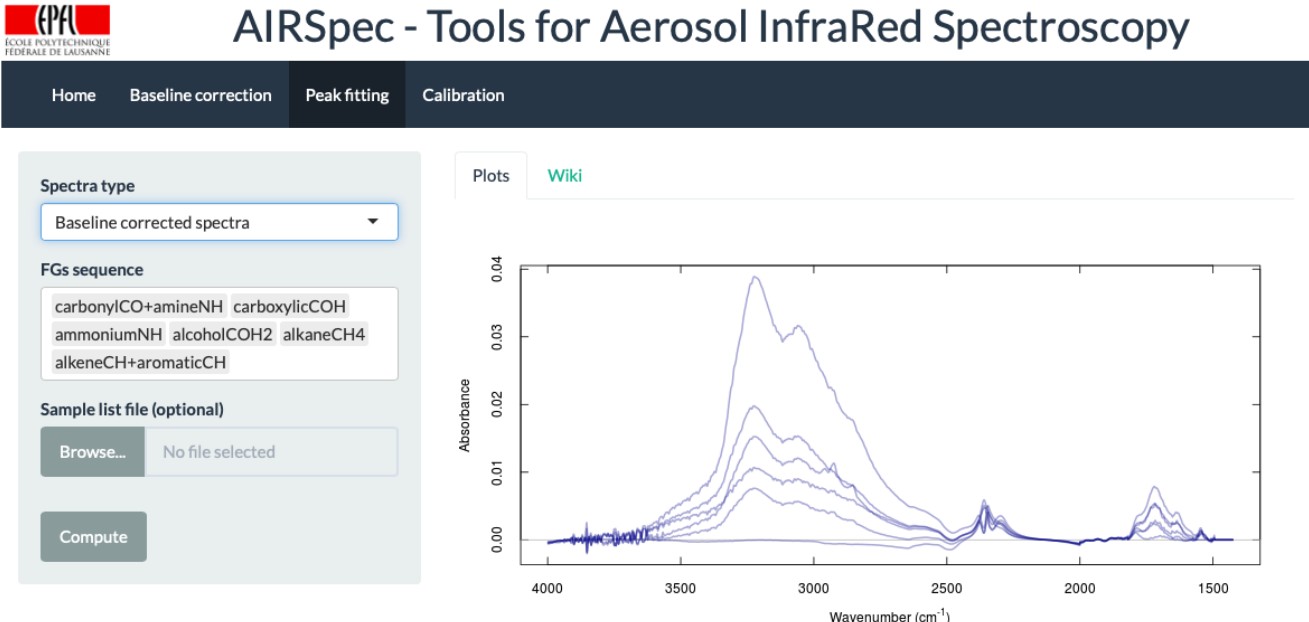

**Figure 6.** AIRSpec peak fitting package tab.





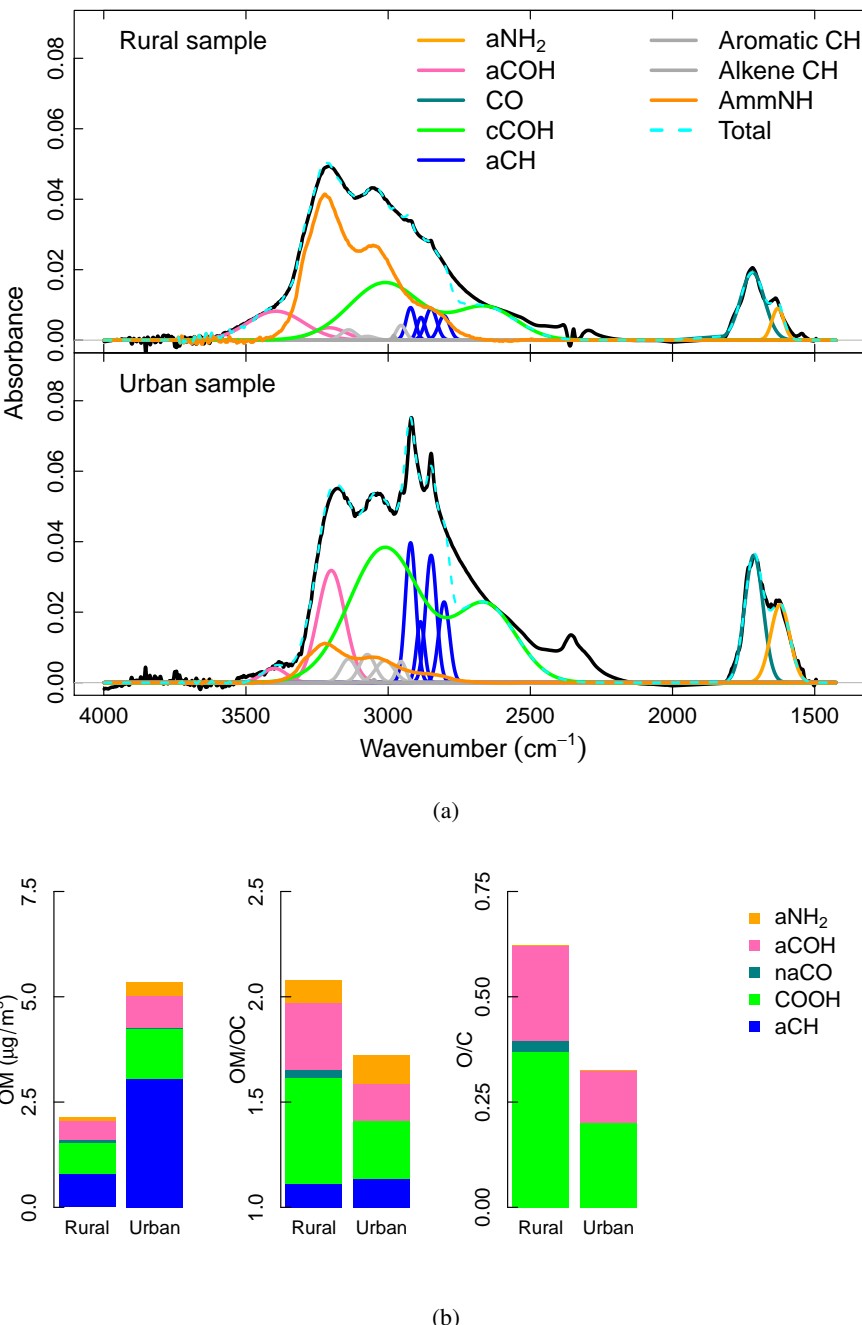

(a)

(b)

**Figure 7.** Peak-fitting package outputs. Example of FG analysis by FT-IR spectroscopy of atmospheric aerosol collected on one Teflon filter. (a) Baseline corrected spectra (black lines) with fitted FGs (colored lines) for the rural (top) and urban (bottom) samples. (b) From left to right the bar plots represent the FG contributions to total OM, their non-carbon contributions to the total OM/OC ratio (Takahama and Ruggeri, 2017), and the O/C atomic ratio for rural and urban samples.



**Table 3.** List of outputs of the Peak fitting chemometric package.

| | Peak-fitting package outputs | |
| --- | --- | --- |
| Category | File | Notes |
| R package input file | input_mpf_user.json | Input parameters used in the computation of the multi-peak fitting algorithm. |
| Package outputs | OM.csv | Table with the estimated organic FG abundances for each sample. The sum of each row gives the OM of each sample. |
| | OM_OC_ratio.csv | Table with the estimated abundances of heteroatoms of OM/OC for each sample. The sum of each row provides the OM/OC for each sample. |
| | O_C_ratio.csv | Table with the estimated O/C ratio for each sample. |
| | moles.csv | Table with the estimated micromoles of functional groups in each sample. |
| Summary files | pkareas.csv | Table with the total areas of the fitted peaks for each functional group and each sample. |
| | fitpars.csv | Parameters of the fitted peaks. |
| Figure | specfits.pdf | Plots of the fitted peaks for each spectrum. |

## 3.3 Multivariate calibration

### 3.3.1 Background

Multivariate calibration is a general technique that can be applied for prediction of functional groups or any arbitrary property to which spectra can be related. Many decisions for this implementation are based on the work of Dillner and Takahama (2015a, b) to predict carbonaceous content and EC from FTIR spectra. The calibration package uses partial least squares regression (PLSR, Wold et al., 1983b; Geladi and Kowalski, 1986) implemented by the PLS package (Mevik and Wehrens, 2007) of the R statistical environment (R Core Team, 2016). The goal is to estimate a set of coefficients $\mathbf{B}$ from a matrix of mean-centered spectra $\mathbf{X}$ for mean-centered response variables $\mathbf{Y}$, with residuals $\mathbf{E}$:

$$\mathbf{Y} = \mathbf{XB} + \mathbf{E}. \tag{9}$$

Because strong correlation of absorbances across wavenumbers (collinearity) exists among variables in $\mathbf{X}$ and the number of wavenumbers in $\mathbf{X}$ exceeds the number of observations, PLSR is used to combine correlated features into a smaller number of latent variables. PLSR performs a bilinear decomposition of both $\mathbf{X}$ and $\mathbf{Y}$: $\mathbf{X}$ is decomposed into a product of orthogonal factors (X-loadings, $\mathbf{P}$) and their respective contributions (scores, $\mathbf{T}$), while $\mathbf{Y}$ is decomposed reconstructed through $\mathbf{T}$ and



their coefficients (or Y-loadings, $\mathbf{Q}$):

$$\mathbf{X} = \mathbf{T}\mathbf{P}^T + \mathbf{F} \tag{10}$$

$$\mathbf{Y} = \mathbf{T}\mathbf{Q}^T + \mathbf{E} \tag{11}$$

$\mathbf{T}$ captures the variations across both $\mathbf{X}$ and $\mathbf{Y}$. $\mathbf{B}$ can be estimated from a matrix of direction vectors $\mathbf{R} = \mathbf{W}\left(\mathbf{P}^T\mathbf{W}\right)^{-1}$

found to maximize covariance of transformed $\mathbf{X}$ with $\mathbf{Y}$ (hat over symbols denote statistically estimated quantities):

$$\hat{\mathbf{T}} = \mathbf{X}\hat{\mathbf{R}}$$

$$\hat{\mathbf{B}} = \hat{\mathbf{R}}\hat{\mathbf{Q}}^T .$$

Candidate models for calibration are generated by varying the number of factors (or latent variables LVs) used to represent the matrix of spectra. The anticipated performance of the model for each number of factors is estimated using root mean square

error of cross-validation (CV) (Hastie et al., 2009; Arlot and Celisse, 2010), and by default, the model with the minimum value is selected.

  Further measures for model interpretation are provided. The explained variation in the response variable, $EV_{rk}^{(Y)}$, varies between 0 and 1 and describes the contribution of component $k$ to the variance of the response variable $r$. The explained variation in the spectra, $EV_{jk}^{(X)}$, varies between 0 and 1 describes the contribution of component $k$ to the variance of the spectra

at wavenumber $j$. The Variable Importance in Projection (VIP) (Wold et al., 1983a) provides complementary information to $EV_{jk}^{(X)}$ in that it assesses the weighting of the $j$-th wavenumber toward the explained variation in the response variable. VIP scores greater than unity for any wavenumber suggests their importance, but this value can in practice vary according to the noise level and the number of uninformative variables (Chong and Jun, 2005a). These expressions are calculated from the measurement and model sum-of-squares:

$$EV_{rk}^{(Y)} = \frac{\mathrm{Tr}\left(\mathbf{T}^T\mathbf{T}\right) \cdot \mathrm{Tr}\left(\mathbf{Q}\mathbf{Q}^T\right)}{\mathrm{Tr}\left(\mathbf{Y}^T\mathbf{Y}\right)} = \frac{\left(\boldsymbol{t}_k^T\boldsymbol{t}_k\right) \cdot \left(\boldsymbol{q}_r^T\boldsymbol{q}_r\right)}{\mathbf{y}_r^T\mathbf{y}_r}$$

$$EV_{jk}^{(X)} = \frac{\mathrm{Tr}\left(\mathbf{T}^T\mathbf{T}\right) \cdot \mathrm{Tr}\left(\mathbf{P}\mathbf{P}^T\right)}{\mathrm{Tr}\left(\mathbf{X}^T\mathbf{X}\right)} = \frac{\left(\boldsymbol{t}_k^T\boldsymbol{t}_k\right) \cdot \left(\boldsymbol{p}_j^T\boldsymbol{p}_j\right)}{\boldsymbol{x}_j^T\boldsymbol{x}_j}$$

$$VIP_{rjk} = \left(N\frac{\sum_{\ell=1}^{k} EV_{r\ell}^{(Y)} w_{j\ell}^2}{\sum_{\ell=1}^{k} EV_{r\ell}^{(Y)}}\right)^{1/2} .$$

$N$ is the number of wavenumbers, and $w$ is the elements of unit normal weight vectors, which together with X-loadings, construct the direction vectors $\mathbf{R}$. $\mathrm{Tr}(\cdot)$ is the trace of the matrix. Their use is described by our companion paper (Reggente

et al., 2018).

### 3.3.2 Implementation

**Browser Interface**. In the calibration tab (Fig. 8), the user can choose to use the uploaded spectra matrix file (uploaded in home tab) or, if already computed, the baseline corrected spectra matrix (output of the baseline correction tab). Two additional





files are required: one containing the response values (target variables), and one containing the list of samples to be used for calibration and test. After providing the response file, the user chooses the target variables of the regression from a list given by the column names of the response file uploaded. The user can choose single or multiple variables, and accordingly to the type of PLS desired (PLS type field). In the case of multiple variables, AIRSpec will process a regression model for each variable

(PLS1) or a regression model for the whole matrix of variables (PLS2). The user can also upload an optional file to use only specified wavenumbers, or to exclude responses below the minimum detection limit of the response variable. Moreover, the user can change the default parameters used in the regression (e.g., fitting algorithm, parameters optimization criteria, limit the number of latent variables). By clicking the `Compute` button, AIRSpec will execute prepared scripts, passing as input a file with the desired configuration. Evaluation of models by RMSE of cross-validation, detailed statistics (figures of merit) of the

calibration models, prediction values, and the diagnostic measures (EVs and VIP) are provided. A list of the available files can be found in the Table 4 and in the dedicated wiki tab in the web interface.

    **Example output**. We present an example of organic and elemental carbon (OC and EC, respectively) prediction analyzed by thermal optical reflectance (TOR). Dillner and Takahama (2015a, b) demonstrated that the FTIR spectra of aerosol samples collected on Teflon filters could be used to estimate OC and EC concentrations by building a calibration model using FTIR

spectra paired with collocated quartz fiber filters analyzed for TOR OC and EC. These models achieved accuracy and precision on a par with the TOR precision for samples collected in the same year and sites as those included in the calibration. Reggente et al. (2016) showed that the same calibration model could be used in different year and sites when concentration range and composition of carbonaceous samples in the calibration set approximately resemble those in the prediction set (new samples for which predictions are desired).

Figure 9 shows the results of the calibration models for OC (top row) and EC (bottom row). 794 IMPROVE samples, and 54 blanks are divided into two sets: one is used for model training and parameter selection (calibration set), and one for the evaluation (test set). The calibration set contains two third of the total (chronologically stratified within each site), and the test set the remaining third. We used the baseline corrected spectra (output of the baseline correction package, Sec. 3.1.2). The first column of Fig. 9a shows the RMSE in cross-validation against the number of components (latent variables), and the dotted

vertical line indicates the number of components selected according to the minimum RMSE. The second and third columns show the scatter plots of predicted against observed (or reference) values for the calibration (b) and test datasets(c) respectively. Bias (median difference between measured and predicted), error (median absolute bias), normalized error (median of the error divided by each measured value) and the coefficient of determination of the linear regression fit of the predicted and measured values ($R^2$) are reported in each scatterplot. The scatterplots and metrics revealed that there is a good agreement between

measured (OC and EC reference) and predicted OC and EC values. A detailed description and discussion of these results are described by Dillner and Takahama (2015a, b). In the evaluation tab (Fig 10), there are interactive plots to highlight the desired spectra. The user can select samples from the scatter plots (using a brush tool, square boxes in Fig 10), and then, the selected spectra will change color (red for calibration samples and blue for test samples). This plot can help the user in interpreting the prediction performances, by, for example, comparing the spectra of samples with different prediction quality.





**Figure 8.** AIRSpec calibration package tab.





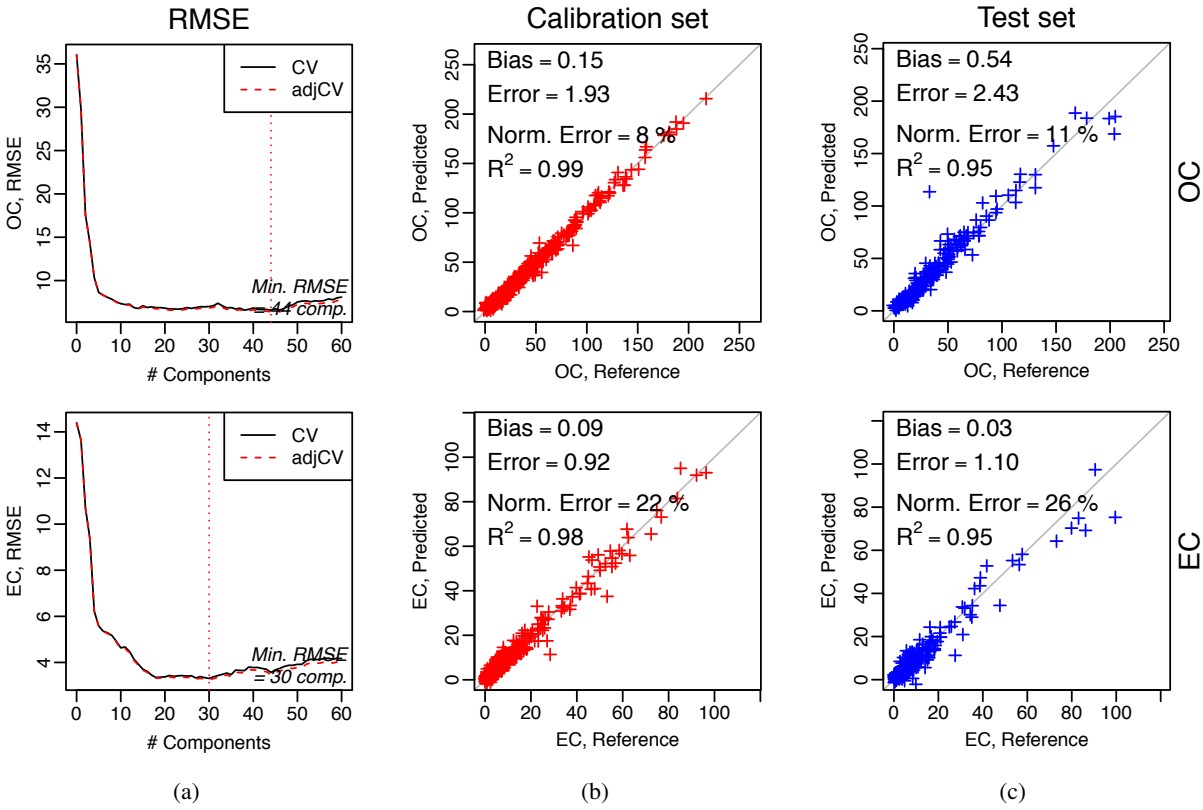

(a)            (b)            (c)

**Figure 9.** Calibration package. Example of calibration output. (a) RMSE in cross-validation against the number of components (latent variables), and the dotted vertical line indicates the number of components selected according to the minimum RMSE. (b) and (c) scatter plots of predicted against observed (or reference) values for the calibration and test datasets respectively.





**Figure 10.** AIRSpec calibration package tab.





**Table 4.** List of outputs of the Calibration chemometric package.

| Calibration package outputs | | |
| --- | --- | --- |
| Category | File | Notes |
| R package input file | `input_mvr_user.json` | Input parameters used in the computation of the calibration model. |
| Package outputs | `fits.rda` | File containing the fitted model for each variable. The file contains all the model specifications (e.g., regression coefficients, scores, loadings). |
| | `pred.rda` | File containing the predicted values for each variable and the whole set of components. |
| | `rmsep.rda` | File containing the RMSE values obtained in cross-validation. |
| | `prediction_table.csv` | Table with the predicted and observed values (for each variable) for the optimal number of components (LVs). |
| Summary file | `stats_table.csv` | File containing the statistics of the fitted models. |
| Figures | `fitsplot.pdf` | Figure with different plots. Each row corresponds to a variable. From left to right, in the first panel, there is the RMSE in cross-validation against the number of components (latent variables), the dotted vertical line shows the number of components selected according to the minimum RMSE. The second and third panels show the scatter plots of predicted against observed (or reference) values for the calibration and test sets. |
| | `{variable}_VIPscores.pdf` | A plot of the Variance Importance in Projection (VIP) scores as described in (Chong and Jun, 2005b). |
| | `{variable}_EVx.pdf` | A plot of the explained variation in X ($EV_{jk}^{(X)}$). |
| | `{variable}_EVy.pdf` | A plot of the explained variation in Y ($EV_{jk}^{(Y)}$). |
| | `{variable}_coefficients.pdf` | A plot of the regression coefficients. |

## 4 Summary and future development

FTIR spectroscopy is a useful tool for obtaining chemical composition of atmospheric PM. However, the complexity of FTIR spectra of PM requires algorithms for consistent interpretation applied across diverse samples. AIRSpec provides a framework for centralizing and disseminating such algorithms. We present three examples of packages implemented for specific tasks:

5  baseline correction, peak fitting, and multivariate calibration. The decoupling of the user interface with the bulk of underlying computation provides flexibility in that exploratory work can be performed with the former, while batch computations can be carried out directly through shell scripts and new scripts can be written to take advantage of existing functions in the underlying packages. The browser interface generates input files so that provenance between input parameters and computation results are preserved, and users can use input files as templates for new computations. The outputs of the program include diagnostic

10  plots, tables of calculations, and statistics that are relevant to the atmospheric aerosol analysis. The modular architecture



exploits common patterns in input specification, computation, and user interaction such that implementation of new collections of algorithms is facilitated by reuse of existing functions. Incorporation of factor and cluster analyses, sparse calibration, and other algorithms are anticipated for future development.

## 5  Code availability

5  Code and software associated with baseline correction, peak fitting, multivariate calibration, and this work are licensed under the GNU Public License v3 and can be downloaded from the following repositories:

- – Basic objects and I/O: https://gitlab.com/aprl/APRLspec

- – Baseline correction: https://gitlab.com/aprl/APRLssb

- – Peak fitting: https://gitlab.com/aprl/APRLmpf

10  – Multivariate calibration: https://gitlab.com/aprl/APRLmvr

- – User interface: https://gitlab.com/aprl/AIRSpec

Instructions are included in the README.md file in each repository. The corresponding author can be contacted for more information.

## 6  Data availability

15  The spectra used for these examples will be made publicly available in the IMPROVE network database.

*Competing interests.*  The authors declare that they have no conflict of interest.

*Acknowledgements.*  The authors thank the National Park Service (cooperative agreement P11AC91045), IMPROVE monitoring network team, and Ann Dillner for the use of their data, and funding for this work from EPFL.





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
