# Peer review of "An open platform for Aerosol InfraRed Spectroscopy analysis – AIRSpec"

_Atmospheric Measurement Techniques, 2018_

## Referee Comment (RC1) · Yang (Referee) · 2 Jan 2019

This paper outlines a new platform to study, develop and share chemometric analysis of functional groups for characterization of atmospheric aerosols from measured FTIR spectra. Three chemometric packages available in the software platform have been listed and detailed. Input and output formats have been listed. Examples of calibration and data analysis has been included in the paper. Overall, this paper describes a software platform where baseline correction of FTIR spectra involving aerosol in measurement path can be made, and functional groups of the particulate matter can be identified. More chemometric analysis can be incorporated by the user. The software package and analysis method followed have been well established and such a comprehensive platform is much desired. The article fits well for publication in AMT

considering the intended audience. It is recommended that the article be accepted for publication. Editorial comments: The article is very well written. Although a couple of typographical errors were noticed. For example: In Section 4. Summary and future development: line 1 "for" is not necessary. "FTIR spectroscopy for is a useful tool for obtaining chemical composition of atmospheric PM"
* * *

---

## Referee Comment (RC2) · Anonymous Referee #1 · 15 Jan 2019

General comments:

This paper reported an open platform for aerosol FTIR (Fourier Transform Infrared) spectra analysis. It is potentially interesting to the community who uses FTIR method to identify organic aerosol sources. As the methods and the data used for demonstration have been already published, I think this paper is an integration of authors' previous works, and I will not comment on the technique of these methods have been used. However, in general, the quantification of the ambient aerosols by using FTIR depends on the functional group (FG) of each component; I encourage the authors to give more discussion of the FG, or give some introduction of the database that readers can access.

Following is some suggestions may improve the paper.

[Figure]

Specific comments:

1, Mid-infrared spectra can be acquired with different methods, such as absorption, reflection, or solvent extraction of filters. Is the authors' platform suitable for all of these spectral methods or only valid for absorption spectra? If it is the last, I suggest the author focuses their title and discussion on absorption spectra.

2, A lot of screenshot of the platform makes the paper looks like a user manual. I suggest the authors to keep only one or two important figures in the paper.

3, In the baseline correction section, figure 5. Is it absolute absorption? What is the method the authors used to get the absorption spectra? In absorption method, the baseline should be treated with careful. Otherwise, it will cause wrong absorption.

4, Fig. 4, it is not clear about segment 1 and segment 2, also there is a misspelling in the figure.

5, Fig. 5, it is not clear about the corresponding functional group of each peak in the figure.

6, Fig. 7, the peaking-fitting package using the decomposed Gaussian peaks for the fitting, how to treat the unknown absorption and overlapping absorption bands?

7, Page 23, the first sentence. "FTIR spectroscopy is a useful tool for obtaining chemical composition of atmospheric PM." It better gives some definition of the chemical composition. Furthermore, an introduction of the potential users is important. What is the progress made with the new platform with the community? It only simplified the data analysis. Is there any improvement on the accuracy?
* * *

---

## Referee Comment (RC3) · Anonymous Referee #2 · 24 Jan 2019

The authors present an open source analysis tool to determine the functional group and concentration of aerosol samples collected on filters. The authors provide details about how the tool analyzes baseline corrections, functional groups, and organic/elemental carbon concentrations with real world examples and comparisons. Though the paper reads as a manual, and I agree with referee#1 that more discussion of functional group quantification would improve the paper, the paper is mostly well written and suitable for publication in AMT after some minor revisions, listed below (and with the possible inclusion on further quantification analysis of functional groups).

Specific comments: 1. Similar to referee#1, I was wondering the suitability of this tool for various methods of IR Spectroscopy analysis, including the use of different filters to collect the samples than the ones analyzed here. It would be beneficial to have a

paragraph to a section describing the feasibility of this tool for other IR techniques.

2. Figure 1 vs Figure 3. It seems that Figure 1 and 3 are the same, but 3 has more detail than 1. Is it necessary to have both figures?

3. Similar to referee#1, I think there are too many screen shots of the website in the main manuscript. If you place them into a supplemental information, it would still be useful for the reader without having too many figures that may make the flow of the paper difficult.

General comments: There are some typos or suggestions in improving the flow of the paper, including: 1. Page 1, Line 8 - 9 (and other locations throughout paper). When the name of the functional group and then the functional group is listed, it would be easier to read it like alcohol (COH) instead of alcohol COH. Also, it should only be defined once in the paper, and use the short hand for the rest of the paper.

2. Page 2, Line 35. Replace semi-colon with and.

3. Page 8, Line 18. St. Marks, Florida instead of St. Marks (Florida).

4. Page 8, Line 28-29. Hard to tell if you have too many penalty in the sentence.

5. Page 10, Line 27. EDF should be capitalized for consistency.

6. Page 11, Fig. 5 caption: Please refer to Fig. 4 so the read knows where segment one and two are coming from.

7. Page 14, Line 7. Space between period and end of sentence

8. Page 14, Line 8. No period at the end of sentence

9. Page 14, Line 20. Are the units supposed to be umoles/cm2? Slightly confused and wanted clarification, as you had just stated how you converted everything to ug/m3, and the values in Fig. 7 are ug/m3.

10. Page 14, Line 25. OM/OC is a ratio, not mass. Not sure why there is mass after

OM/OC.

---

## Author Comment (AC1) · 25 Mar 2019

**Response to reviewer comments: "An open platform for Aerosol InfraRed Spectroscopy analysis – AIRSpec"**

We thank the editor and reviewers for feedback on the manuscript. We have addressed each comment below with responses in blue font, and submitted revised manuscript (with new additions highlighted in red font and old text for deletion in gray).

**Anonymous Referee #1**

**General comments**  This paper reported an open platform for aerosol FTIR (Fourier Transform Infrared) spectra analysis. It is potentially interesting to the community who uses FTIR method to identify organic aerosol sources. As the methods and the data used for demonstration have been already published, I think this paper is an integration of authors' previous works, and I will not comment on the technique of these methods have been used. However, in general, the quantification of the ambient aerosols by using FTIR depends on the functional group (FG) of each component; I encourage the authors to give more discussion of the FG, or give some introduction of the database that readers can access.

We thank the reviewer for the positive feedback. Moreover, since this work is the companion paper of the article https://doi.org/10.5194/amt-2018-331 that focuses on methods to quantify FGs, we did not include an exhaustive discussion of the FG quantification.

We have added the following sentence (Section 3.2.1), to provide the reference to access the article:

"A more detailed discussion of FG quantification in atmospheric aerosol samples by FTIR is provided by a companion paper of this work (Reggente et al., 2018)."

**Specific comments**  Following is some suggestions may improve the paper.

1 Mid-infrared spectra can be acquired with different methods, such as absorption, reflection, or solvent extraction of filters. Is the authors' platform suitable for all of these spectral methods or only valid for absorption spectra? If it is the last, I suggest the author focuses their title and discussion on absorption spectra.

While our experience is primarily with transmission mode analysis on particles collected on PTFE filters, we have also applied this code to spectra obtained with solvent-extracted solutions deposited on ZnSe crystals (Arangio et al., Applied Spectroscopy, in press, 2018). Furthermore, independent implementations of the peak fitting algorithm described here has been applied to ATR-FTIR spectra (Faber et al. 2017) and solvent-extracted spectra using DRIFTS (Chen et al., 2016). We have added the following text to the manuscript:

"The examples shown are provided for absorbance spectra acquired from transmission mode analysis. In principle, the methods described [baseline correction, peak-fitting, and calibration algorithms] are applicable to spectra that can be converted to equivalent absorbance spectra — for instance, measurements from attenuated total reflectance (ATR) and diffuse reflectance Fourier Transform spectroscopy (DRIFTS) can be converted to approximate absorbance spectra using the weak-absorption approximation (Harrick, 1979) or Kubelka-Munk theory (Kubelka and Munk, 1931), respectively."

We have additionally modified references in the text which might suggest that the analysis is only limited to particles collected on PTFE and analyzed by transmission mode analysis. Moreover, as new chemometric packages can be easily added using the modular structure of AIRSpec, we would like to keep the title general to indicate that it is an "open platform".

References:

Arangio, A., Delval, C.E.L., Ruggeri, G., Dudani, N., Yazdani, A.: Electrospray film deposition for solvent-elimination infrared spectroscopy, Applied Spectroscopy, in press, 2018.

Chen, Q., Ikemori, F., Higo, H., Asakawa, D., and Mochida, M.: Chemical Structural Characteristics of HULIS and Other Fractionated Organic Matter in Urban Aerosols: Results from Mass Spectral and FT-IR Analysis, Environmental Science & Technology, 50, 1721– 1730, doi:10.1021/acs.est.5b05277, 2016.

Faber, P., Drewnick, F., Bierl, R., and Borrmann, S.: Complementary online aerosol mass spectrometry and offline FT-IR spectroscopy measurements: Prospects and challenges for the analysis of anthropogenic aerosol particle emissions, Atmospheric Environment, 166, 92–98, doi:10.1016/j.atmosenv.2017.07.014, 2017.

Harrick, N. J.: Internal Reflection Spectroscopy, Harrick Scientific Corp., 1979.

Kubelka, P. and Munk, F.: Ein beitrag zur optik der farbanstriche, Zeitschrift für technische Physik, 12, 593–601, 1931.

2 A lot of screenshot of the platform makes the paper looks like a user manual. I suggest the authors to keep only one or two important figures in the paper.

We agree with the referee, and we have decided to keep two Figures of the platform: (i) the screenshot of the AIRSpec homepage (Figure 2) because it is the first page that the user encounter, and (ii) the screenshot of AIRSpec calibration package tab (old Figure 10) because it shows the interactive plots functionality, that we have added.

3 In the baseline correction section, figure 5. Is it absolute absorption? What is the method the authors used to get the absorption spectra?

We thank the reviewer for pointing out this possible missatement. While Beer-Lambert law strictly applies to pure absorption spectra, our baseline corrected spectra is only an approximation of absorption spectra in that gross changes in spectra due to drift and scattering obtained from the ratio of single-beam spectra (of sample and background) are removed with the smoothing spline. We have revised the introductory paragraph of the baseline correction section (Section 3.1) to read:

"Instrumental drift and scattering by particles and substrate used for their collection create interfering signals that hinder quantitative analysis. For the transmission mode absorbance spectra of particles collected onto PTFE presented here (obtained by ratioing single beam spectra of sample to background), the subtraction of the spectrum filter before sample exposure (blank) from the aerosol spectrum has been shown to provide an insufficient remedy (Takahama et al., 2013)."

The old Figure 5 now is Figure 4 because we have removed the old Figure 4 as suggested by the referee at above (point 2). In Figure 4, the absorbance is in arbitrary unit (A.U.), which is now stated in the Figure caption. The spectra in Figure 4 refer to aerosol collected on Teflon filters and measured using the transmission method.

4 In absorption method, the baseline should be treated with careful. Otherwise, it will cause wrong absorption.

We are aware that the baseline correction needs to be treated carefully, and therefore the AIRSpec implements a chemometric package that has been already reviewed and published as reported in the text.

5 Fig. 4, it is not clear about segment 1 and segment 2, also there is a misspelling in the figure.

We have removed Figure 4 as suggested by the referee in a previous comment (point 2).

6 Fig. 5, it is not clear about the corresponding functional group of each peak in the figure.

The absorption profiles and the wavenumbers where each functional group (FG) absorbs are shown in Figure 5 (old Figure 7).

7 Fig. 7, the peaking-fitting package using the decomposed Gaussian peaks for the fitting, how to treat the unknown absorption and overlapping absorption bands?

The peak-fitting package aims to find the best fit to the baseline corrected measured spectra of the overlapping Gaussian peaks and line shapes which represent the absorption profile of individual functional groups. Referring to Figure 5 (old Figure 7) the spectra given by the sum of the fitted peaks (fitted spectra) are the dashed cyan lines. The divergences between the fitted spectra and the baseline corrected spectra may due to (i) the baseline correction, or (ii) to the presence of chemical species not included in the analysis. For example, the peak around 2200 $cm^{-1}$ is due to the presence of $CO_2$ in the measurement chamber.

8 Page 23, the first sentence. "FTIR spectroscopy is a useful tool for obtaining chemical composition of atmospheric PM." It better gives some definition of the chemical composition.

We have rephrased the sentence as follow (Section 4, page 21 lines 2–3): "FTIR spectroscopy is a useful tool for obtaining the functional group representation of the chemical composition of atmospheric PM."

9 Furthermore, an introduction of the potential users is important. What is the progress made with the new platform with the community? It only simplified the data analysis. Is there any improvement on the accuracy?

We have added the following sentence in the Abstract (page 1, lines 19–21):

"Moreover, AIRSpec facilitates the exploratory work, can be used by FTIR spectra acquired with different methods, and can be extended easily with new chemometric packages when they will be available. Therefore AIRSpec provides a framework for centralizing and disseminating such algorithms."

**Anonymous Referee #2**

The authors present an open source analysis tool to determine the functional group and concentration of aerosol samples collected on filters. The authors provide details about how the tool analyzes baseline corrections, functional groups, and organic/elemental carbon concentrations with real world examples and comparisons. Though the paper reads as a manual, and I agree with referee#1 that more discussion of functional group quantification would improve the paper, the paper is mostly well written and suitable for publication in AMT after some minor revisions, listed below (and with the possible inclusion on further quantification analysis of functional groups).

We thank the reviewer for the positive feedback. Moreover, since this work is the companion paper of the article https://doi.org/10.5194/amt-2018-331, that focuses on methods to quantify FGs, we did not include an exhaustive discussion of the FG quantification.

We have added the following sentence (Section 3.2.1), to provide the reference to access the article:

"A more detailed discussion of FG quantification in atmospheric aerosol samples by FTIR is provided by a companion paper of this work (Reggente et al., 2018)."

**Specific comments**

1 Similar to referee#1, I was wondering the suitability of this tool for various methods of IR Spectroscopy analysis, including the use of different filters to collect the samples than the ones analyzed here. It would be beneficial to have a paragraph to a section describing the feasibility of this tool for other IR techniques.

We have added the following paragraph in the text (Section 1) also in response to comment #1 of Referee #1:

"The examples shown are provided for absorbance spectra acquired from transmission mode analysis. In principle, the methods described [baseline correction, peak-fitting, and calibration algorithms] are applicable to spectra that can be converted to equivalent absorbance spectra — for instance, measurements from attenuated total reflectance (ATR) and diffuse reflectance Fourier Transform spectroscopy (DRIFTS) can be converted to approximate absorbance spectra using the weak-absorption approximation (Harrick, 1979) or Kubelka-Munk theory (Kubelka and Munk, 1931), respectively."

2 Figure 1 vs Figure 3. It seems that Figure 1 and 3 are the same, but 3 has more detail than 1. Is it necessary to have both figures?

> We thank the reviewer for pointing out that it is not clear the difference between Figure 1 and Figure 2. Figure 1 represents a diagram of the AIRSpec framework and the chemometric packages implemented and the interaction between them. At the moment, the calibration and peak fitting packages uses the output of the baseline correction package. We also show that additional packages can be added (denoted by dotted box and arrows). Figure 3, on the other hand, represents the modular structure of a single chemometric package. We changed the captions of these two figures to explain better what they represent.

> > Fig.1: AIRSpec architecture diagram and workflow. Solid boxes and arrows refer to chemometric packages implemented in the current version. Dotted box and arrows denotes that it is possible incorporate new packages in the modular framework.

> > Fig.3: Chemometric package diagram and workflow. Solid boxes and arrows refer to shiny modules used by each chemometric package (necessary modules). Supplementary features are handled by ad-hoc modules and denoted by dotted boxes and arrows.

3 Similar to referee#1, I think there are too many screen shots of the website in the main manuscript. If you place them into a supplemental information, it would still be useful for the reader without having too many figures that may make the flow of the paper difficult.

> We agree with the referees, and we have decided to keep two Figures of the platform: (i) the screenshot of the AIRSpec homepage (Figure 2) because it is the first page that the user encounter, and (ii) the screenshot of AIRSpec calibration package tab (old Figure 10) because it shows the interactive plots functionality, that we have added.

**General comments**   There are some typos or suggestions in improving the flow of the paper, including:

1 Page 1, Line 8 - 9 (and other locations throughout paper). When the name of the functional group and then the functional group is listed, it would be easier to read it like alcohol (COH) instead of alcohol COH. Also, it should only be defined once in the paper, and use the short hand for the rest of the paper.

> We thank the referee for the suggestion. We have defined the acronyms in the abstract (page 1, lines 7–8), and then defined them again the first time they are used in the body of the paper (page 2, line 35).

2 Page 2, Line 35. Replace semi-colon with and.

> Corrected

3 Page 8, Line 18. St. Marks, Florida instead of St. Marks (Florida).

> Corrected

4 Page 8, Line 28-29. Hard to tell if you have too many penalty in the sentence.

> Corrected

5 Page 10, Line 27. EDF should be capitalized for consistency.

> Corrected

6 Page 11, Fig. 5 caption: Please refer to Fig. 4 so the read knows where segment one and two are coming from.

> Since we have removed the Figure 4, as suggested by referee #1 (specific comments, item 2) and referee #2 (specific comments, item 3), we have added in the caption of Fig. 4 (old Fig. 5) the following sentence:

> "In the current implementation of the chemometric package, Segment 1 and Segment 2 refer to spectra in the regions 4000–1820 cm$^{-1}$ and 2000–1500 cm$^{-1}$, respectively (Kuzmiakova et al., 2016)."

7 Page 14, Line 7. Space between period and end of sentence

Corrected

8  Page 14, Line 8. No period at the end of sentence

Corrected

9  Page 14, Line 20. Are the units supposed to be umoles/cm2? Slightly confused and wanted clarification, as you had just stated how you converted everything to ug/m3, and the values in Fig. 7 are ug/m3.

We thank the referees for pointing out the error and ambiguous paragraph. We have rephrased the statement as follows:

"AIRSpec provides FG abundance in several representations: the areal density ($\mu$mole/cm$^2$) calculated from eq. 5, and the areal mass density ($\mu$g/cm$^2$) obtained from eq. 8 and the atomic masses of each element. The OM (in units of $\mu$g/cm$^2$) summed from FG contributions is also provided. To obtain atmospheric concentrations ($\mu$g/m$^3$), it is necessary to multiply these areal mass densities by the substrate collection area and divide by the volume of air sampled (3.53 cm$^2$ and 32.8 m$^3$, respectively, for examples shown in this manuscript)."

10  Page 14, Line 25. OM/OC is a ratio, not mass. Not sure why there is mass after OM/OC.

Corrected

**Referee Huinan Yang**

This paper outlines a new platform to study, develop and share chemometric analysis of functional groups for characterization of atmospheric aerosols from measured FTIR spectra. Three chemometric packages available in the software platform have been listed and detailed. Input and output formats have been listed. Examples of calibra- tion and data analysis has been included in the paper. Overall, this paper describes a software platform where baseline correction of FTIR spectra involving aerosol in measurement path can be made, and functional groups of the particulate matter can be identified. More chemometric analysis can be incorporated by the user. The soft- ware package and analysis method followed have been well established and such a comprehensive platform is much desired. The article fits well for publication in AMT considering the intended audience. It is recommended that the article be accepted for publication.

We thank the reviewer for the positive feedback.

**Editorial comments**   The article is very well written. Although a couple of typographical errors were noticed. For example: In Section 4. Summary and future development: line 1 "for" is not necessary. "FTIR spectroscopy for 
[revised manuscript text omitted]

Checkbox for non-negativity constraint | Sample list file |
| Peak fitting | | Spectra type
Bond sequence | Sample list file |
| Calibration | Response file
Case file | Spectra type
Variables

[revised manuscript text omitted]

---

## Author Comment (AC2) · 25 Mar 2019

The comment was uploaded in the form of a supplement:
https://www.atmos-meas-tech-discuss.net/amt-2018-332/amt-2018-332-AC2-supplement.pdf
* * *

---

## Author Comment (AC3) · 25 Mar 2019

The comment was uploaded in the form of a supplement:
https://www.atmos-meas-tech-discuss.net/amt-2018-332/amt-2018-332-AC3-supplement.pdf
* * *